# Study of Oxidation of Ciprofloxacin and Pefloxacin by ACVA: Identification of Degradation Products by Mass Spectrometry and Bioautographic Evaluation of Antibacterial Activity

Barbara Żuromska-Witek [1], Paweł Żmudzki [2], Marek Szlósarczyk [1], Michał Abram [2], Anna Maślanka [1] and Urszula Hubicka [1,*]

1   Department of Inorganic and Analytical Chemistry, Faculty of Pharmacy, Jagiellonian University Medical College, Medyczna 9, 30-688 Kraków, Poland; barbara.zuromska@uj.edu.pl (B.Ż.-W.); m.szlosarczyk@uj.edu.pl (M.S.); anna.maslanka@uj.edu.pl (A.M.)
2   Department of Medicinal Chemistry, Faculty of Pharmacy, Jagiellonian University Medical College, Medyczna 9, 30-688 Kraków, Poland; pawel.zmudzki@uj.edu.pl (P.Ż.); michal.abram@uj.edu.pl (M.A.)
*   Correspondence: urszula.hubicka@uj.edu.pl; Tel.: +48-12-620-5480

**Abstract:** The new RP-HPLC-DAD method for the determination of ciprofloxacin and pefloxacin, next to their degradation products after the oxidation reaction with 4,4′-azobis(4-cyanopentanoic acid) (ACVA) was developed. The method was validated according to the guidelines of the International Council for Harmonization of Technical Requirements for Pharmaceuticals for Human Use (ICH) and meets the acceptance criteria. The experimental data indicate that the course of the oxidation process depends on the type of fluoroquinolone (FQ), the incubation time and temperature. The performed kinetic evaluation allowed us to state that the oxidation of FQs proceeds according to the second-order kinetics. The degradation products of the FQs were identified using the UHPLC-MS/MS method and their structures were proposed. The results obtained by the TLC-direct bioautography technique allowed us to state that the main ciprofloxacin and pefloxacin oxidation products probably retained antibacterial activity against *Escherichia coli*.

**Keywords:** ciprofloxacin; pefloxacin; RP-HPLC-DAD method; oxidation studies; ACVA; degradation products; mass spectrometry; direct bioautography

## 1. Introduction

Fluoroquinolones (FQs) are known as antimicrobial agents with a broad spectrum of activity and are generally well-tolerated antibacterial agents, but are not free from mild, reversible, and more serious side effects [1,2]. In the territory of the European Union (EU), FQs have been subject to several referral procedures, resulting in restrictions on some indications for ciprofloxacin in 2008 [3]. According to the EU/EEA report (2018), the trend in consumption of FQs is decreasing, although some European countries still show "overconsumption" of this group for systemic use in the hospital sector. The report mentioned above also recommends the correct use of these antibiotics, listing particular indications in which FQs should not be used [4].

Among the approved FQs in Europe are pefloxacin (PEF) and ciprofloxacin (CIP), and the latter is also approved by the FDA [2]. In this context, the stability of drugs should also be taken into account as a factor affecting their safety and efficacy. Impurities resulting from the drug degradation process may be responsible for the limited efficacy and occurrence of the side effects; reassuming, and achieving the physicochemical stability of drugs is necessary to ensure their quality and safety [5].

The degradation study of active product ingredients (API) and finished pharmaceutical products is a critical point of pharmacotherapy and quality control of each therapeutic agent, especially in the case of FQs, known to be susceptible to some degradation processes [6].

Stability tests are important tools in the detection, identification, and quantification of possible degradation products that can be realized as accelerated stability, long-term, and intermediate studies [7,8]. Compared to "classical" stability studies, forced degradation studies help shorten the time of the degradation process or obtain a measurable level of degradation products. There are many goals for forced degradation studies, such as to discover the susceptibility of API to degradation processes such as hydrolysis, oxidation, thermolysis, or photolysis, to determine the intrinsic stability of a drug substance in a formulation, or to develop the stability-indicating analytical methods. The mentioned studies are also essential in understanding the chemical properties of drug molecules to solve stability-related problems or invent more stable formulations [9].

The aspect of the fate of drugs and their metabolites in the environment is also substantial, as in the human body; moreover, the processes of oxidative degradation of drugs are very important and are widely used in environmental protection. It is well known that FQs, as emerging contaminants, occur in sewage, surface, and groundwater; consequently, they have toxic effects on aquatic organisms and may increase the number of multi-drug-resistant bacteria. Oxidation reactions are one of the key ways to eliminate drugs from the environment [6,10,11]. Advanced oxidation processes used particularly to remove toxic pollutants that occur in water and sewage include chemical processes (oxidation with $O_3$ and $H_2O_2$, Fenton's reaction) [12,13] and photochemical processes (UV photolysis, processes using $UV/H_2O_2$, $UV/O_3$, $UV/H_2O_2/O_3$, or photo-Fenton reaction) [14], leading to the formation of hydroxyl radicals ($OH^\bullet$). The hydroxyl radical is one of the most reactive non-selective oxidants, with a quite high redox potential of approximately 2.8 V [15].

Forced oxidative degradation studies of pharmaceuticals using a variety of oxidizing conditions and agents are routinely performed in drug stability studies in the pharmaceutical industry due to the key role of oxidation processes in drug degradation [16–18]. The mechanism of oxidation is based on autoxidation or the formation of free radicals, which are involved in the propagation of radical-initiated chain reactions. The selected stress conditions should simulate the decomposition of drug products during processes such as manufacturing, storage or confectioning but there are no detailed regulatory guidelines on the physical conditions or specific oxidizing agents to be used [9]. There are no recommended oxidants to carry out the oxidation process, but many species are widely used for this purpose, such as hydrogen peroxide, oxygen, metal ions, and azo radical initiators [8,19]. In the case of the last mentioned, this forced oxidative degradation model allows reproducible results to be achieved and is a good source of information on how to carry out forced degradation studies [20]. Additionally, the application of azo radical initiators such as ACVA was described as a good predictive model of the main oxidative degradation products observed in pharmaceutical formulations [19]. The thermal decomposition of ACVA leading to the formation of two cyanoalkyl radicals and then peroxide radicals oxidizing the tested compounds is shown in Scheme 1.

The selection of an oxidizing agent, its concentration, and the conditions of the planned degradation studies depend on the structure of the drug molecule. The mechanism of the oxidative reaction of API involves an electron transfer to form reactive ions [9]. The susceptible to electron transfer are compounds with, i.e., amino, and phenolic groups to give N-oxides and hydroxylamine. The functional group with labile hydrogen-like benzylic carbon, allylic carbon, and tertiary carbon or $\alpha$-positions with respect to the heteroatom is susceptible to oxidation to form hydroperoxides, hydroxide, or ketone [9,18,21]. The oxidation products can show different activity than the drug substance (higher toxicity, lower efficacy), which in some cases may compromise therapeutic action [22,23]. The antibacterial activity of FQs (CIP, ofloxacin) depends on the structure of the core quinolone and the presence of the carboxylic and carbonyl groups at positions 3 and 4, but significant transformations of the other functional groups can decrease antibacterial activity [24].

**Scheme 1.** Thermal decomposition of ACVA and radical formation.

One of the techniques for evaluating the antibacterial activity of drug degradation products is bioautography: a technique employing a suitable chromatographic process followed by a biological detection system [25,26]. Thin-layer chromatography direct bioautography (TLC–DB) is a screening technique that is used to study the biological properties, especially the antibacterial and antifungal properties, of various substances. In TLC–DB, the developed TLC plate is sprayed or immersed in a suspension of microorganisms growing in a liquid medium and then incubated in a humid atmosphere. Zones of inhibition of bacterial growth arise in places where there are antibacterial substances on the TLC plates. In order to locate and develop antibacterial substances, the obtained bioautogram is sprayed most often with an aqueous solution of a tetrazolium salt (e.g., 3-(4,5-dimethyl-2-thiazolyl)-2,5-diphenyl-2H-tetrazolium bromide, MTT). Next, the MMT is converted by dehydrogenase, a mitochondrial enzyme of living bacteria, to purple formazan. Compounds with antibacterial properties, placed on the TLC plate, inhibit the growth of bacteria and creme white zones of inhibition of microbial growth are formed on a coloured background [27].

The stability of selected FQs has been extensively tested, particularly according to their photosensitivity and as potential emerging contaminants [27–29]. Our previous studies on CIP photodegradation were performed in the solid state [30] and in the presence of excipients [31], photocatalysts [28], or under oxidation conditions [32]. CIP was reported to be quite stable in 5% dextrose in water or 0.9% sodium chloride [33], but susceptible to degradation in acidic solution and the presence of metal ions [34]; furthermore, CIP was found to be susceptible to oxidative degradation after the influence of potassium permanganate in an acidic environment [35]. Less scientific research was focused on PEF degradation, but in a similar context: photocatalytic degradation [28,36], degradation by $KMnO_4$ of a strong oxidant in acidic condition [35] or by in-situ generated oxygen-based oxidation agents [37]. Continuing our previous research on the oxidative stability of FQs [38], we decided to extend it to more stable compounds from this group, such as CIP and PEF. CIP and PEF belong to different generations of FQs; they have a similar lipophilicity, but they differ slightly in the substituents in the 1 and 7 positions of the quinolone core. The PEF has an ethyl group at position 1 and a 4-methylpiperazin-1-yl group at position 7. Whereas CIP has at position 1 cyclopropyl group and piperazin-1-yl group at position 7. Next to the similar structure CIP and PEF are common pharmacotherapeutic agents occurring in the same types: solid (oral tablets) and liquid (the ear or eye drops) pharmaceutical dosage forms, which enabled us to perform parallel decomposition studies. As previously mentioned, the oxidation processes of FQs have been the subject of several dozen publications, but the aim of the described research was to obtain the maximum decomposition of the tested compounds to eliminate FQs from the natural aquatic environment; however, the literature

on the stability of FQs lacks research on their oxidation in radical reactions with azonitrile radicals useful in pharmaceutical sciences. To the best of our knowledge, the kinetics of this reaction have not been studied, and the degradation mechanisms of CIP and PEF have not been reported. We believe that an understanding of the fundamentals of the free radical chemistry reaction of these FQs will enhance the development of strategies to control and target selective chemical transformation for pharmaceutical applications.

Thus, the aim of this study was to evaluate the oxidation process of CIP and PEF under the influence of the azo initiator of radical reactions of ACVA. In order to achieve the above goals, a new, sensitive RP-HPLC-DAD method allowing for the determination of CIP and PEF with its oxidation products was developed and validated. The proposed HPLC method enables the separation of tested FQs from their degradation products and fulfils the acceptance criteria for the stability-indicating analytical method. The kinetic evaluation of the oxidation process was performed, and the degradation product structures were proposed using the UHPLC-MS/MS method. In addition, the antimicrobial activity of the main oxidation products of FQs was tested against the *E. coli* strain using a fast, simple, and low-cost TLC-direct bioautography technique. It seems that the studies carried out have a cognitive and practical aspect. In the available literature, the degradation pathways of CIP and PEF under the influence of ACVA have never been reported, and we have not found comprehensive research on this subject. There were also no reports in the literature on the use of TLC-direct bioautography to assess the antibacterial activity of the resulting degradation products of FQs.

## 2. Materials and Methods

### 2.1. Chemicals and Reagents

All the chemicals and reagents were purchased from commercial suppliers: Ciprofloxacin hydrochloride (Cat. No. LRAB3671); Pefloxacin mesylate dihydrate (Cat. No. SLBC5834V), Triton X-100, Thiazolyl blue tetrazolium bromide (MTT), Mueller-Hinton bullion, agarose for microbiology, 4,4′-azobis(4-cyanopentanoic acid) (ACVA), hydrogen peroxide (30%), anhydrous sodium dihydrogen phosphate, *o*-phosphoric acid (85%) and HPLC grade water were from Merck Life Science (Darmstadt, Germany) while methanol and acetonitrile (HPLC grade) were from WITKO (Łódź, Poland).

### 2.2. Standard Solution

The tested FQs in the amount of 0.1000 g were weighed with an analytical balance and dissolved in a volume of 100 mL of methanol. For the validation of the method, solutions in the concentration range 0.03–0.18 mg·mL$^{-1}$ for CIP and 0.03–0.17 mg·mL$^{-1}$ for PEF were prepared.

### 2.3. Preparation of Samples and Execution of Oxidation Tests with the Participation of ACVA

The test samples were prepared by making 25 mL of a 0.5 mM solution of FQ in a 10 mM water-acetonitrile (50:50 *v/v*) solution of ACVA. Then, the ready solution was transferred to the 2.0 mL vials for the incubation in a dry block heating system (QBH$_2$ Grant, Grant Instruments) at digitally controlled temperatures of 40 °C, 50 °C and 60 °C. Additionally, ACVA control samples that did not contain FQ and control samples of fluoroquinolone without ACVA solution were prepared [38]. The samples of CIP were incubated at 40 °C for 336 h; at 50 °C for 72 h and at 60 °C for 24 h. The PEF samples were incubated at 40 °C for 336 h, at 50 °C for 96 h and at 60 °C for 60 h. At appropriate time intervals, which were mainly dependent on the type of FQ and temperature, the content of one vial was analysed in duplicate.

### 2.4. Preparation of Samples and Execution of Oxidation Tests with the Participation of 3% $H_2O_2$

The test samples were prepared by making 25 mL of a 0.5 mM FQ solution in a solution of 3% $H_2O_2$. Then the procedure described in 4.3 above was followed and the samples were

incubated at room temperature for 56 days. At a weekly time interval, the content of the prepared vial of the suitable FQ was analysed. The analyses were performed in duplicate.

### 2.5. RP-HPLC-DAD Analysis

The HPLC system, HITACHI, High-Technologies Corporation (Tokyo, Japan) equipped with an autosampler (L-2200) and a photodiode array detector (L-2455) was used. The chromatographic analysis of CIP and PEF was performed on a reversed phase ACE 5 C18-PFP column, Advanced Chromatography Technologies Ltd. (Aberdeen, Scotland) (250 × 4.6 mm, 5 μm particle size), coupled with a guard column. The column temperature was 30 °C. Chromatographic separation was achieved using a mixture of 0.025 M phosphate buffer (pH = 3.20 adjusted with o-phosphoric acid) and acetonitrile in a gradient elution procedure (Table 1). The flow rate of the mobile phase was 1.0 mL·min$^{-1}$, and the injection volume was 10 μL. The analysis time was 25 min. A five-min time interval was used between injections. The chromatograms for both quinolones were registered at 277 nm.

**Table 1.** Gradient elution procedure.

| Time (Min) | Buffer (%) | Acetonitrile (%) |
|:---:|:---:|:---:|
| 0 | 80 | 20 |
| 5 | 80 | 20 |
| 10 | 50 | 50 |
| 15 | 30 | 70 |
| 20 | 30 | 70 |
| 25 | 80 | 20 |

### 2.6. Method Validation

The method for determining CIP and PEF in the presence of oxidation products was validated according to the ICH Q1A (R2) guidelines [39]. The specificity of the presented HPLC method was assessed by the analysis of 0.5 mM solutions of CIP and PEF after reaction with ACVA at 50 °C and 48 h of incubation.

To determine the linearity for CIP and PEF, the solutions in the concentration range of 0.03–0.18 mg·mL$^{-1}$ for CIP and 0.03–0.17 mg·mL$^{-1}$ for PEF were prepared. The results were obtained using Statistica 13.3 software package (TIBCO Software Inc., Palo Alto, CA, USA) with all requirements. Based on the ICH [39] guidelines, LOD and LOQ for the tested compounds were calculated as follows: $LOD = 3.3 \times S_e/a$ and $LOQ = 10 \times S_e/a$, where $S_e$ is the standard error of estimation and a is the slope of the curve calibration. The values of LOD and LOQ were then experimentally confirmed.

The precision of the method for CIP and PEF was investigated by analysing 6 individually prepared solutions with a concentration of 0.10 mg·mL$^{-1}$ (100% level).

The same procedure was followed for another day by a different analyst to check the intermediate precision of the described methods. The results were then elaborated by the calculated RSD (%) value of the peak area of tested FQs. The robustness of the method was checked by intentionally making small modifications to the flow rate, the temperature of the column, and the pH value. The flow rate was changed to 1.10 and 0.90 mL·min$^{-1}$ (primary flow rate was 1.0 mL·min$^{-1}$). The temperature of the column to 28 °C and 32 °C (instead of 30 °C) and the pH of the mobile phase ±5% from the initial value was modified.

### 2.7. Determination of Kinetic Parameters of the Oxidation Process

To assume the order of the reaction, ln c or 1/c dependence curves were plotted against the incubation time. The fit to the linear model was examined and the correlation coefficients were determined. The symbol c represents the percentage of remaining undegraded FQ. Subsequently, the reaction rate constants (k) and the degradation times of 50% for the tested FQs ($t_{0.5}$) were calculated.

### 2.8. UHPLC/MS/MS Analysis

The UHPLC-MS/MS analysis was performed as described in the publication [38]. Detailed conditions for the UHPLC-MS/MS analysis are also provided in the Supplementary Materials.

### 2.9. Thin-Layer Chromatography–Direct Bioautography (TLC–DB)

2.9.1. TLC Conditions

Precoated silica gel 60F$_{254}$, 1 mm PLC glass plates 20 cm × 20 cm (no 1.13895.0001 Merck) were chosen as the stationary phase. A mixture of chloroform, methanol, and ammonia 25% (43:43:14, *v/v/v*) was used as the mobile phase.

Standard solutions, sample solutions after oxidation with ACVA, and ACVA solution were applied on the plates using the Linomat 5 applicator (CAMAG, Muttenz, Switzerland). The concentrations of the standard solutions were selected at a level, which inhibits the growth of *E. coli*. The solutions of FQs after the oxidation reaction were used in amounts that allowed to obtain the peak areas for the major degradation products comparable to the peak areas of the FQs standards.

The above-mentioned solutions were applied on the plates as 1 cm wide bands in the following amount:

- Track 1—60 μL of standard solution of CIP (0.18 mg mL$^{-1}$);
- Tracks 2–4—160, 180, and 200 μL of CIP sample after oxidation reaction with ACVA at 60 °C for 24 h;
- Track 5—60 μL of standard solution of PEF (0.17 mg mL$^{-1}$);
- Tracks 6–8—200, 225, and 250 μL of PEF sample after the oxidation reaction with ACVA at 60 °C for 60 h;
- Track 9—60 μL of ACVA solution after incubation at 60 °C for 60 h.

The plates were developed in a glass chamber (29.5 × 26 × 10, Sigma-Aldrich, Saint Louis, MO, USA) and were removed when the solvent front has moved to 19.5 cm from the original sample position and subsequently allowed to dry.

2.9.2. Cultivation of Test Bacteria

The test strain of *Escherichia coli* (ATCC 25922) was purchased from the American Type Culture Collections. The bacterial suspension used for the TLC–DB method was prepared according to the procedure developed and optimized by Choma et al. [26]. Details on the procedure are provided below.

The bacterial strain in a vial with the glycerol solution was incubated at 37 °C for 5 min to revive the culture. 0.1 mL of the contents of the vial were withdrawn and placed on the Petri dishes by spreading it with a sponge over the surface of the MacConkey agar and then placed in an incubator at 30–35 °C. After 24 h from the obtained growth, the microorganisms were collected with a loop and streaked onto a new aliquot of MacConkey agar and then returned to the incubator for 24 h at 30–35 °C.

*E. coli* colonies were collected using a loop and transferred to tubes with a physiological saline solution. The density of the suspension was measured using a densitometer until a turbidity of not less than 7 on the McFarland scale was obtained.

Colonies of *E. coli* were put into 250 mL Mueller-Hinton broth (pH 7.2 ± 0.2, adjusted with Hepes) and preincubated at 37 °C for 20 h (the bacterial mean concentration equalled 4.8 × 10$^8$ CFU/mL). Next, 50 mL of bacterial suspension obtained directly from the preincubation was added to 1 L Mueller-Hinton broth (pH 7.2 ± 0.2, 0.05% agarose) and incubated at 37 °C for 3 h. After this time, the bacterial mean concentration equalled 4.6 × 10$^7$ CFU/mL.

2.9.3. TLC–DB Analysis

The TLC plates, after complete evaporation of the mobile phase, were immersed for 5 min in the bacterial suspension in the glass vessel. Subsequently, they were placed in

a new vessel and incubated at 37 °C for 5 h. To visualise zones of inhibition of bacterial growth, the plates were sprayed with 0.2% MTT aqueous solution and again incubated at 37 °C for 0.5 h. The entire procedure was repeated 3 times.

## 3. Results

### 3.1. Optimization of Chromatographic Conditions and Method Validation

To separate CIP and PEF from their oxidation products, two chromatographic columns were tested as stationary phases: ACE-5 C18-PFP (250 × 4.60 mm) and Kinetex 5u XB-C18 100A (250 × 4.60 mm, particle size 5 μm core-shell type) Phenomenex. The solutions of the tested FQs after 48 h of incubation at 50 °C in 10 mM ACVA were applied. The Kinetex 5u XB-C18 A column was used as the first stationary phase, while the mobile phase was a mixture of 0.05 M phosphate buffer at pH = 3.20 and acetonitrile in various proportions (80:20, 85:15, 87:13, *v/v*). No satisfactory separation of FQs and their degradation products was obtained. The stationary phase was then replaced with the ACE-5 C18-PFP column. The analyses were repeated, but under the applied conditions, no satisfactory separation of the tested substances was achieved. Finally, it was found that to develop the best method for the oxidation study of selected FQs, a mixture of phosphate buffer (pH = 3.20) and acetonitrile in gradient elution should be used (Table 1).

The proposed RP-HPLC-DAD method meets the acceptance criteria for the specificity. In the developed chromatographic conditions, well-shaped and separated peaks were obtained for examined FQs and oxidation products. The resolution factors between the FQ peaks and the nearest degradation products were $R_{sCIP} > 1.78$ and $R_{sPEF} > 1.02$. The determined asymmetry factors ($A_s$) were 1.19 and 1.25 for CIP and PEF, respectively. At the selected analytical wavelength of 277 nm, the peaks of the ACVA control solution were absent and did not interfere with the peaks of degradation products (Figure S1).

The peak visible in the chromatogram with a retention time of about 20 min is the "ghost" peak that has been formed while the gradient parameters have been inversed to the initial conditions. The chromatograms of the compounds before the oxidation reaction are shown below (Figures 1 and 2). The peak marked as "z" ($t_R \approx 13.38$ min) registered on the chromatogram of the standard solution of PEF is an impurity that occurs in the purchased standard (content about 0.55%).

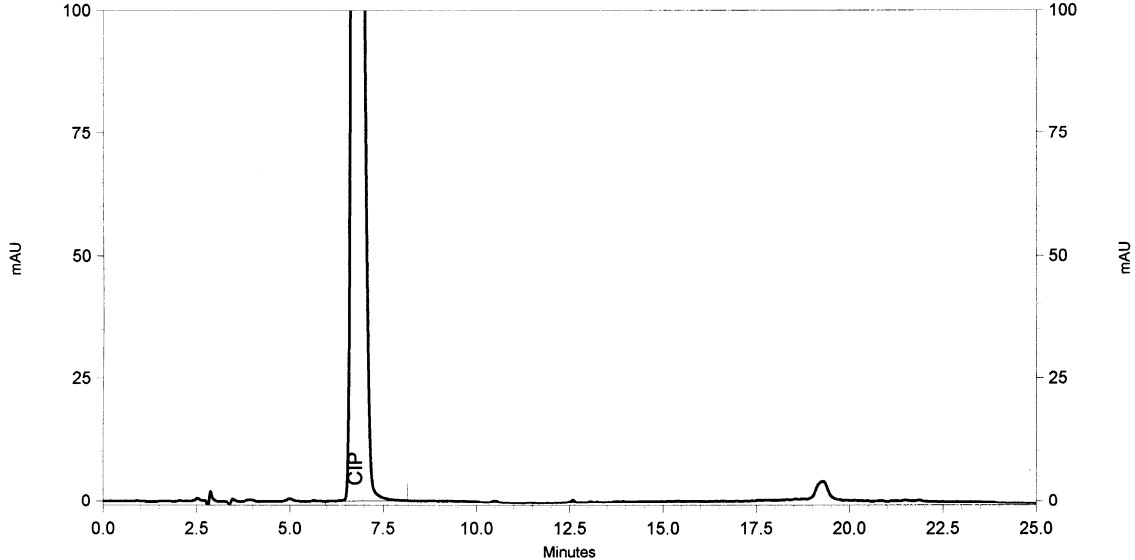

**Figure 1.** Chromatogram of the CIP standard solution at a concentration of 0.5 mM recorded at 277 nm.

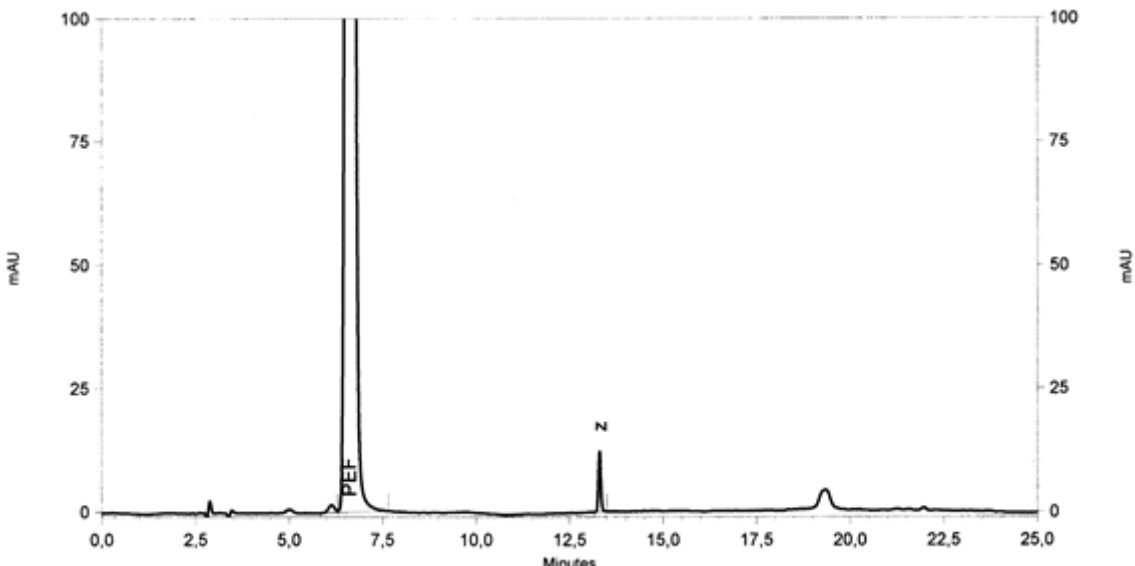

**Figure 2.** Chromatogram of PEF standard solution at a concentration of 0.5 mM recorded at 277 nm.

The regression analysis of the response as a function of FQs concentration was performed using the least-squares method, based on which it was found that the linear model is valid in the tested concentrations range. The determination coefficients were greater than 0.99 and the y-intercepts of the linear equation for CIP and PEF were statistically insignificant. The distribution of the residuals is normal according to the result of the Shapiro-Wilk normality test ($p > 0.05$). The linearity range was obtained in the following concentration ranges 0.03–0.18 mg·mL$^{-1}$ for CIP and 0.04–0.17 mg·mL$^{-1}$ for PEF. Table 2 shows all the regression analysis results obtained.

**Table 2.** Method validation summary.

| Parameter | CIP | PEF |
|---|---|---|
| $t_R$ (min) [a] | 6.89 | 7.03 |
| LOD (mg·mL$^{-1}$) | 0.008 | 0.01 |
| LOQ (mg·mL$^{-1}$) | 0.02 | 0.04 |
| Linearity range (mg·mL$^{-1}$) | 0.03–0.18 | 0.04–0.17 |
| Regression equation (y): Slope ($a \pm S_a$) | $2438 \times 10^5 \pm 3{,}351{,}895$ | $2217 \times 10^5 \pm 655{,}508$ |
| Intercept ($b \pm S_b$) | $-858 \times 10^3 \pm 390{,}850$ | $-149 \times 10^4 \pm 5{,}697{,}105$ |
| $t = b/S_b$ | $-2.20 < t_{\alpha,f}$ | $-2.27 < t_{\alpha,f}$ |
| Normality of residuals (Shapiro-Wilk test) [b] | 0.9243 ($p = 0.39$) | 0.8519 ($p = 0.06$) |
| Determination coefficient $r^2$ | 0.9985 | 0.9974 |
| Precision (RSD%) | 1.73 | 1.10 |
| Intermediate precision (RSD%) | 1.87 | 1.37 |

[a] Mean $\pm$ SD ($n = 6$); LOD, LOQ—The limits of detection and quantification, respectively; Regression equation: y = ac + b (y—peak area, c—concentration of solution); $S_a$, $S_b$—the standard deviations of slope and intercept, respectively; t—calculated value of Student's *t*-test (df = 4, $\alpha$ = 0.05; $t_{\alpha,f}$ = 2.776 critical value of Student's *t*-test); [b] normal distribution of residuals if $p > 0.05$.

The method is characterized by high sensitivity. The LOD and LOQ values were found to be 0.008 and 0.02 mg·mL$^{-1}$ for CIP, while 0.01 and 0.04 mg·mL$^{-1}$ for PEF. For precision and intermediate precision, the RSDR% values of less than 1.87% were received (Table 2). An inconsiderable change ($\pm 5\%$ from the optimal value) in the flow rate, the temperature of the column, and the pH of the buffer caused the worse separation of the examined FQs

from the oxidation products. Consequently, it is important to use the developed parameters of the method.

### 3.2. Oxidation Study of Examined FQs

The radical initiator ACVA has been used for the oxidation of CIP and PEF in solutions incubated at three temperatures: 40 °C, 50 °C, and 60 °C. An oxidation survey of CIP and PEF was also performed with hydrogen peroxide at room temperature (RT) commonly used in forced degradation studies.

The results of the experiment indicated that the progress of the oxidation process by using ACVA depends on the type of FQs, incubation time, and temperature. With the extension of the incubation time and the increase in temperature, a faster and greater degree of decomposition of the compounds tested was observed.

The oxidation process of CIP was the fastest at 60 °C, slower at 50 °C, and the slowest at 40 °C. The percentage of CIP decomposition after 336 h of incubation at 40 °C was 37.77%, at 50 °C after only 72 h of incubation it was 18.00%, while at the highest temperature 60 °C after 24 h it was 40.40%. The oxidation of CIP at 40 °C resulted in the formation of three main degradation products with percentages content of 3.95%, 3.83%, 27.80% for the peaks with $t_R \approx 5.07$ min; 13.54 and 15.33 min, respectively. The four other peaks had a small amount (<0.7%) (Figure 3). At 50 °C and 60 °C, one major degradation product of CIP was formed, which content of 15.67% and 35.02% ($t_R \approx 15.33$ min), respectively, and from 4 to 6 oxidation products below 1.20% (Figures 4 and 5).

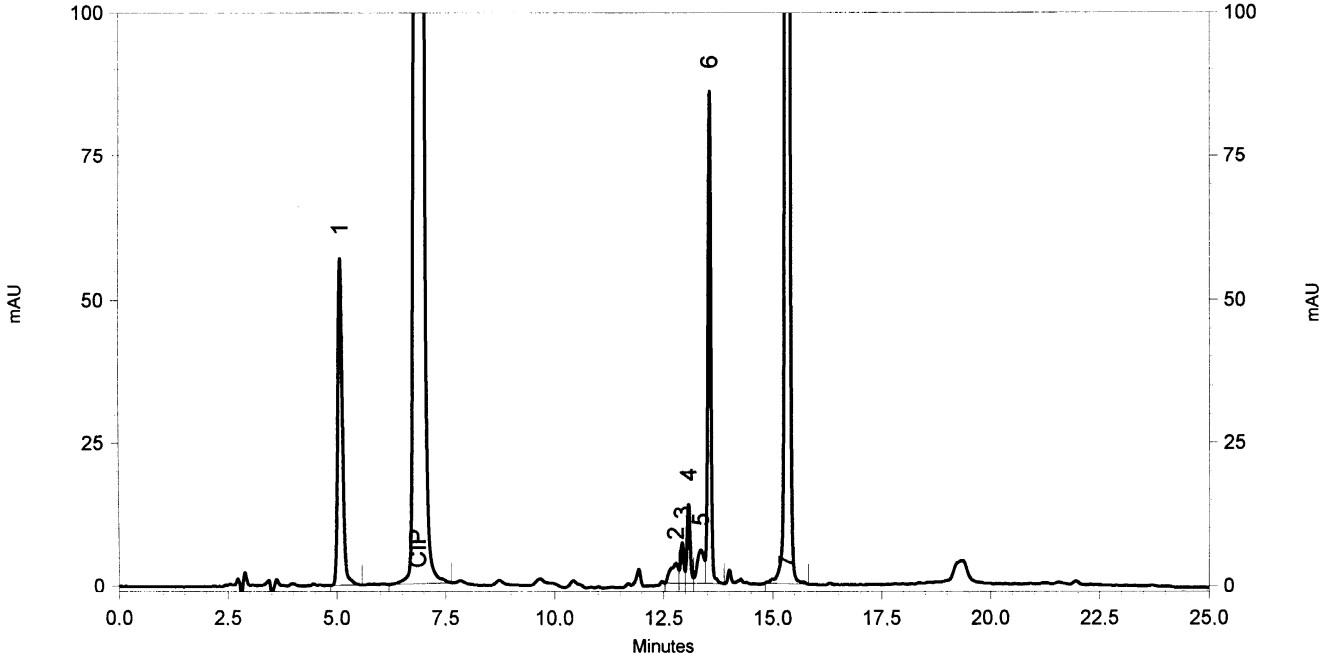

**Figure 3.** Chromatogram of the CIP oxidized by ACVA after incubation for 336 h at 40 °C.

In the case of the oxidation of PEF by ACVA, the results showed that the process causes the formation of one main degradation product with a percentage of above 9.50% ($t_R \approx 7.89$ min), one with an amount of above 1.14% ($t_R \approx 5.00$ min) and the 3 others with a small amount (<0.38%). As with CIP, the degradation of PEF was the slowest at 40 °C and after 336 h of incubation was 14.81%. The process was faster at higher temperatures, at 50 °C and 60 °C the percentage of degradation of PEF was 11.63% (after 96 h of incubation) and 24.67% (after 60 h of incubation), respectively (Figures 6–8).

After 56 days of the incubation of CIP or PEF with $H_2O_2$ at room temperature, no peaks from the degradation products were recorded. It suggests that the degradation process of the tested FQs did not take place under the conditions used. (Figures 9 and 10).

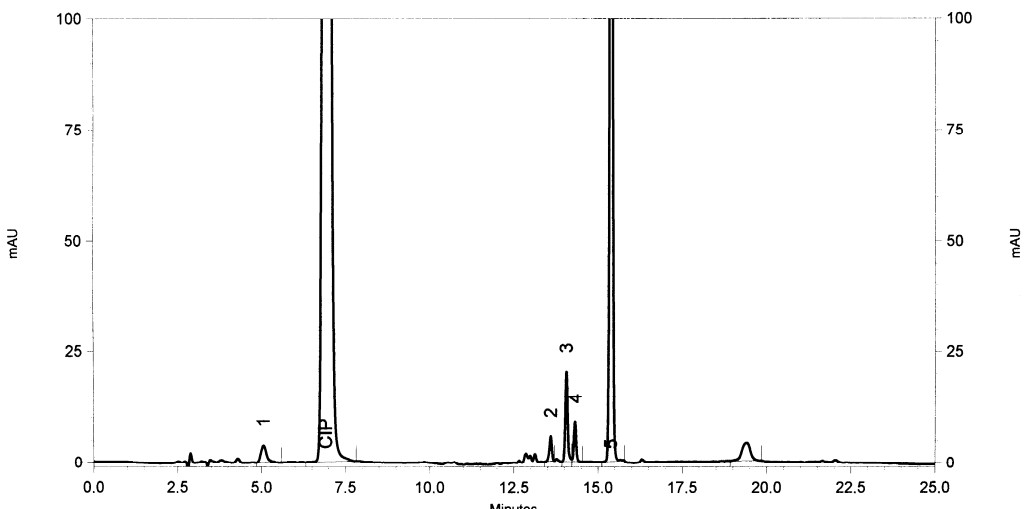

**Figure 4.** Chromatogram of the CIP oxidized by ACVA after incubation for 72 h at 50 °C.

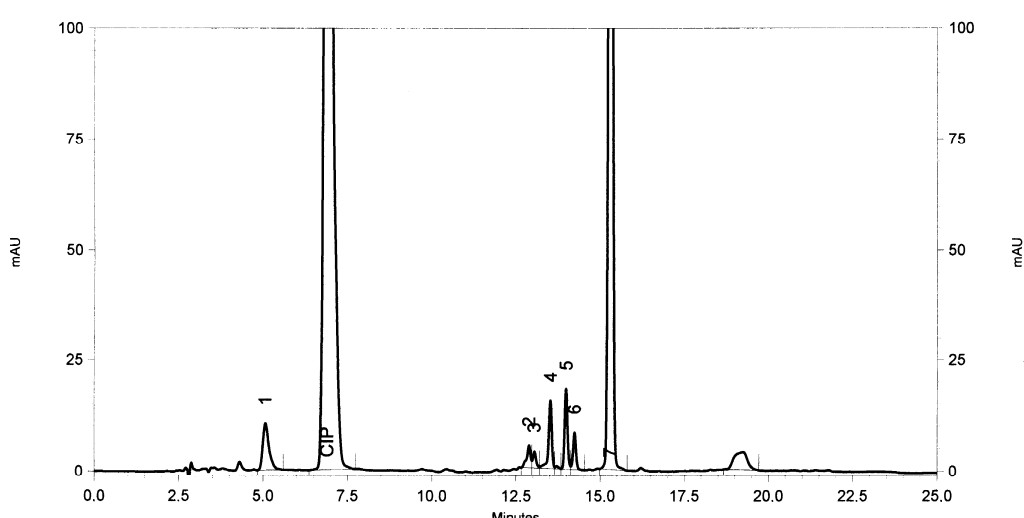

**Figure 5.** Chromatogram of the CIP oxidized by ACVA after incubation for 24 h at 60 °C.

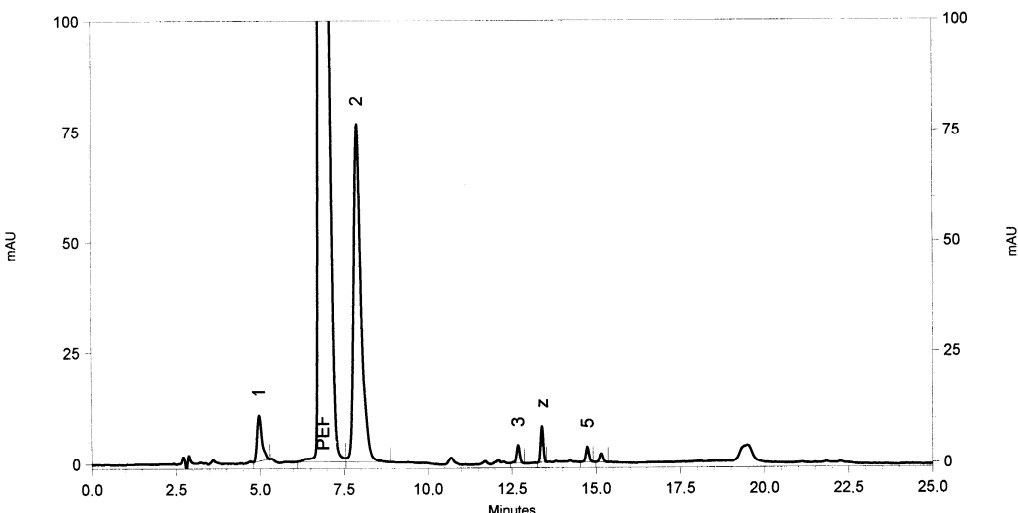

**Figure 6.** Chromatogram of the PEF oxidized by ACVA after incubation for 336 h at 40 °C.

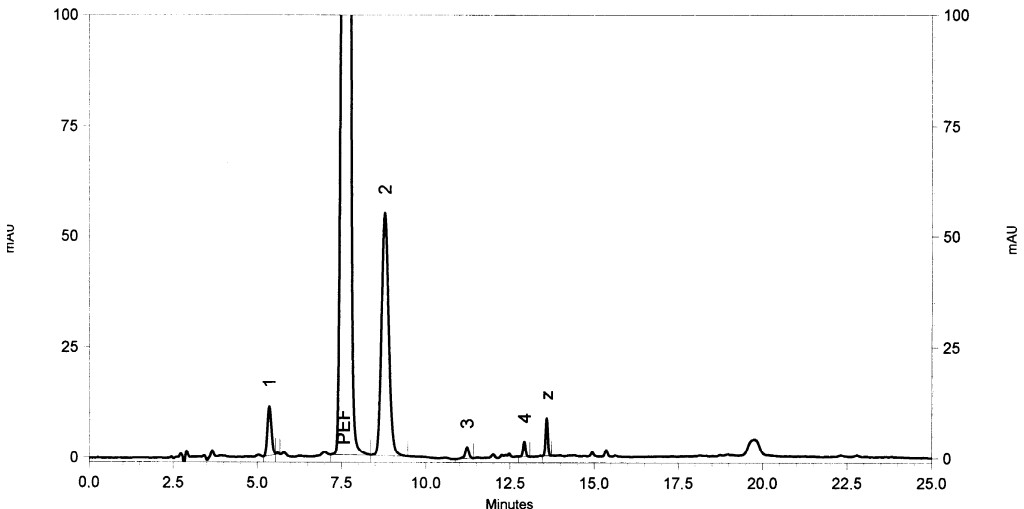

**Figure 7.** Chromatogram of the PEF oxidized by ACVA after incubation for 96 h at 50 °C.

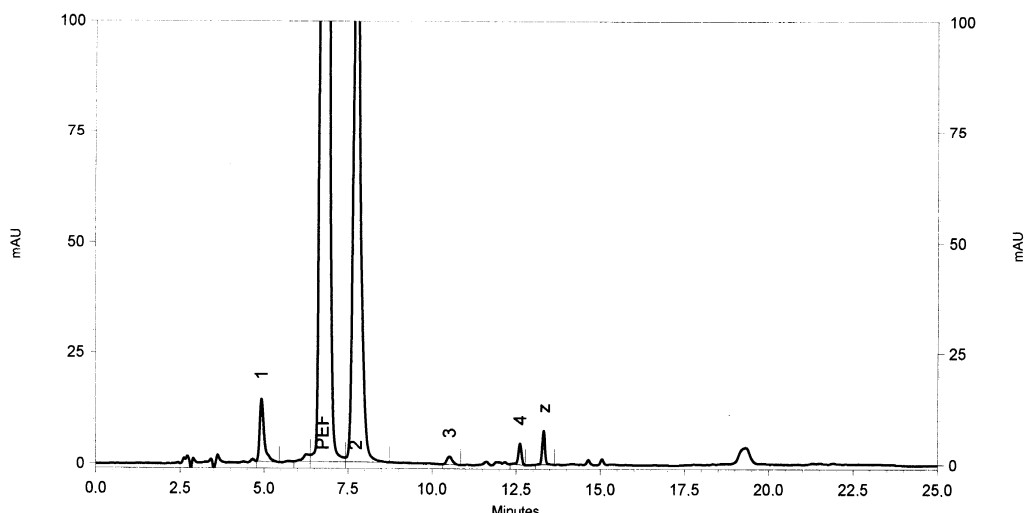

**Figure 8.** Chromatogram of the PEF oxidized by ACVA after incubation for 60 h at 60 °C.

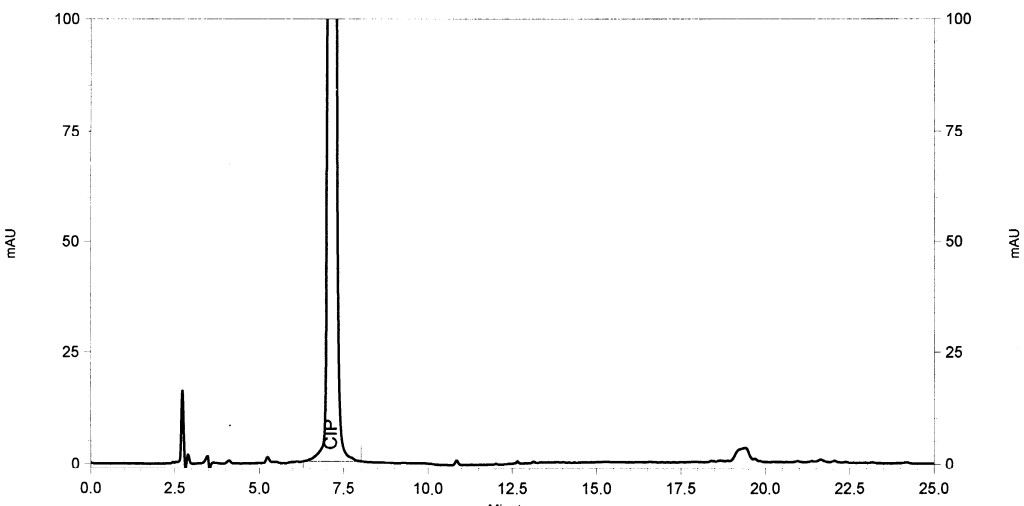

**Figure 9.** Chromatogram of the CIP oxidized by H$_2$O$_2$ after incubation for 56 days at room temperature.

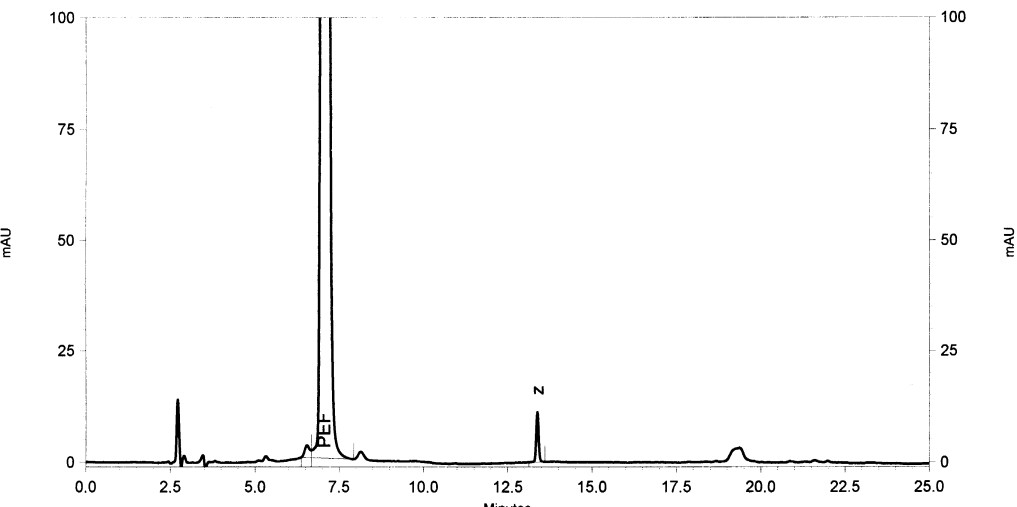

**Figure 10.** Chromatogram of the PEF oxidized by $H_2O_2$ after incubation for 56 days at room temperature.

### 3.3. Kinetic Evaluation

On the basis of a highly significant correlation demonstrated by the analysis of the plots [1/c = f(t)] for both the oxidation of CIP and PEF while using ACVA it was stated that the oxidation process followed the kinetics of second-order (Figure 11); moreover, the degradation rate constant k and the half-life $t_{0.5}$ were calculated for all examined conditions. The obtained results confirmed that the rate of the oxidation process is dependent on the type of FQs tested, and the temperature (Table 3).

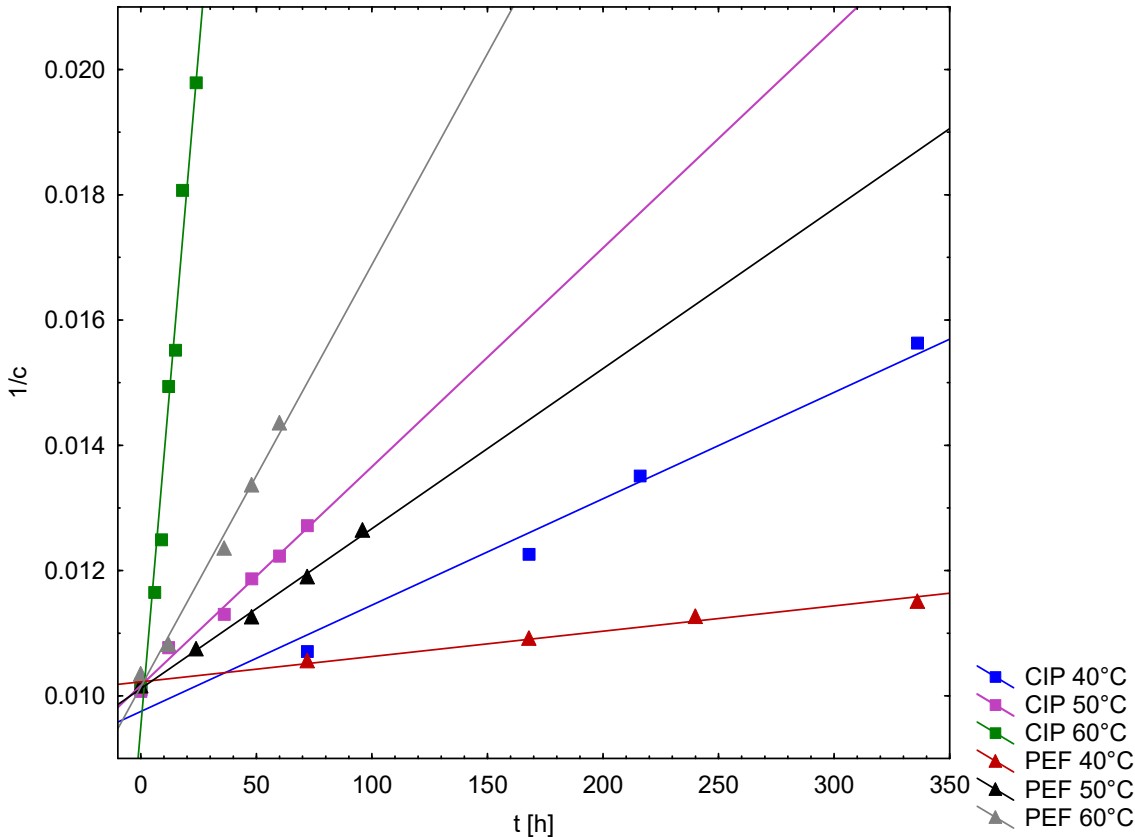

**Figure 11.** The 1/c = f(t) graph of oxidation of CIP and PEF solutions under the influence of ACVA.

**Table 3.** The kinetic results of the CIP and PEF oxidation by ACVA.

| Compound | Temperature | k (h$^{-1}$) | t$_{0.5}$ (h) | r |
|----------|-------------|--------------|---------------|---|
| CIP | 40 °C | $1.70 \times 10^{-5}$ | 588.2 | 0.9912 |
| | 50 °C | $3.49 \times 10^{-5}$ | 286.5 | 0.9935 |
| | 60 °C | $4.00 \times 10^{-4}$ | 25.0 | 0.9834 |
| PEF | 40 °C | $4.04 \times 10^{-6}$ | 2475.2 | 0.9926 |
| | 50 °C | $2.55 \times 10^{-5}$ | 392.2 | 0.9975 |
| | 60 °C | $6.74 \times 10^{-5}$ | 148.4 | 0.9941 |

k—rate constant, r—correlation coefficient.

After analysing the data, it was found that oxidation of CIP and PEF under the influence of ACVA proceeds faster with increasing temperature. At all examined temperatures of the experiment, there were significant differences in the rate of oxidation reaction for CIP and PEF solutions. The estimated kinetic parameters of the CIP oxidation process under the influence of ACVA indicate that, for all tested temperatures, it runs faster than for PEF solutions (Table 3). The oxidation of CIP at 60 °C was about 5.9 times faster than that of PEF. In the case of PEF, oxidation under the influence of ACVA at 40 °C was about 4.2 times slower than for CIP under the influence of ACVA (Table 3).

### 3.4. Identification of Oxidation Products

Based on the UHPLC-MS/MS technique (corresponding chromatograms and MS spectra are shown in Supplementary Materials section Figures S2–S5), the structures of the main oxidation products of the examined FQs were proposed and supported by fragmentation patterns. The suggested structures of the degradation products are shown in Tables 4 and 5.

**Table 4.** Proposed structures of the degradation products of CIP.

| Compound | t$_R$ (Min) | (M + H$^+$) | Fragmentation Ions | Structure |
|----------|-------------|-------------|--------------------|-----------|
| CIP | 2.70 | 332.1 | 231.1, 245.1, 288.2, 314.1 |  |
| CIP-1 | 4.18 | 360.1 | 231.1, 243.1, 273.1, 342.1 |  |
| CIP-2 | 4.94 | 371.1 | 215.1, 231.1, 243.1, 259.1, 325.0, 353.0 |  |

**Table 5.** Proposed structures of the degradation products of PEF.

| Compound | $t_R$ (Min) | (M+H$^+$) | Fragmentation Ions | Structure |
|---|---|---|---|---|
| PEF | 2.67 | 334.2 | 290.2, 316.1 |  |
| PEF-1 | 2.79 | 350.2 | 290.2, 332.1 |  |

The CIP degradation process was found to primarily affect the piperazine moiety, leading to its oxidation and the creation of the CIP-1 product. Further degradation involved cleavage of the piperazine ring at the nitrogen atom of the secondary aliphatic amine group and substitution of fluorine, leading to the main product CIP-2. Fragmentation of all of the observed degradation products involved cleavage of the carboxylic group with further fragmentation of the piperazine moiety. The fragmentation patterns of CIP and its degradation products are shown in Schemes 2–4. The structure of the third CIP oxidation product has not been determined.

**Scheme 2.** Proposed fragmentation pattern of CIP.

The obtained results are confirmed in the literature. Numerous authors indicate the high reactivity of the secondary amine group and the tertiary amine group of the piperazine ring in the oxidation reactions of FQs. Oxidizing compounds including radicals attacking the piperazine ring can lead to cleavage of the C-N bond on the secondary nitrogen atom [40,41] and/or the tertiary aromatic nitrogen atom and the formation of keto derivatives of piperazine [42–44]. In addition, oxidation reactions, especially with the participation of free radicals, can lead to the replacement of the F atom on the quinolone

moiety ring by the hydroxyl group [45–47]. The observed CIP-1 product under oxidation conditions used by us has also been reported for the CIP reaction with potassium permanganate at pH = 7 [43], whereas the second obtained oxidation product CIP-2 and its chemical structure were reported for the first time.

**Scheme 3.** Proposed fragmentation pattern of CIP-1.

**Scheme 4.** Proposed fragmentation pattern of CIP-2.

In an investigation of the oxidation of PEF under the influence of ACVA, it was found that the degradation process affects the carboxyl moiety of PEF. The degradation pattern of the PEF-1 product, involving elimination of water and elimination of $CO_2$, may suggest that the oxidation of the moiety occurred via the addition of the oxygen atom to the double bond of the carbonyl group. Another possible product might be peroxycarboxylic acid, which possesses a similar fragmentation pattern. The fragmentation patterns of PEF and its oxidation product are shown in Schemes 5 and 6. Based on the available literature, it can be concluded that the proposed chemical structure of the PEF degradation product differs from that obtained in several other studies on the oxidation of PEF and other FQs [35–37,48,49]; therefore, we suggest that the obtained degradation product and its proposed chemical structure were reported for the first time.

**Scheme 5.** Proposed fragmentation pattern of PEF.

**Scheme 6.** Proposed fragmentation pattern of PEF-1.

In addition, an Open-Source software Osiris DataWarrior V5.5.0 (Idorsia Pharmaceuticals Ltd., Switzerland) was used to rapidly assess the potential toxicity risks (mutagenicity, carcinogenicity, irritation, and reproductive effect) of the CIP and PEF oxidation products based on their proposed chemical structure. Fortunately, only one of the degradation products (PEF-1) probably shows more toxicity risks than the parent compound. The program

indicates a potential high tumorigenic risk or a medium risk of effect on the reproduction of the PEF-1 product with the structure shown in Table 5; however, if the PEF-1 product has the structure of peroxycarboxylic acid, the program indicates a likely moderate mutagenic effect, a high risk of carcinogenicity and irritation, and a high reproductive effect. The evaluation of the potential adverse biological effects of the degradation products of FQs that may appear in a medicinal product is very important for pharmaceutical technology and the pharmaceutical industry.

*3.5. Evaluation of Antimicrobial Activity of Degradation Products by TLC–DB*

Evaluation of the toxicity and antimicrobial activity of derivatives formed during the degradation processes of pharmacologically active compounds is very important for the pharmaceutical industry and environmental protection. Degradation products may exhibit similar, greater or lesser toxicity and antimicrobial activity than the parent compounds [45,50]. The frequently used methods for assessing the antimicrobial activity of synthetic and natural compounds and their degradation products include the agar disc diffusion method, broth dilution method with visual or spectrophotometric detection, ATP bioluminescence test, and TLC-bioautography method [51]. The latter, among the methods mentioned above, allows for determining the microbiological activity of the parental and derivative compounds after their chromatographic separation; it is advantageous because it gives an answer to which degradation product has retained microbiological activity and illustrates potential changes [45,50].

Our research used a combination of thin-layer chromatography with microbiological detection—direct bioautography—to investigate the antimicrobial properties of the main degradation products derived from the oxidation of CIP and PEF. During the study, the strain of Gram-negative *Escherichia coli* bacteria (ATCC 25922) was used. The selected bacterial strain is a typical representative of its group commonly found in nature; moreover, the tested FQs show antibacterial activity against *E. coli* [52,53].

To separate the FQs and oxidation products, the TLC conditions previously used to analyse CIP photodegradation were successfully applied [30]. Standard solutions of CIF and PEF, solutions of tested FQs after the oxidation reaction, and a solution containing only ACVA were applied to each TLC plate, as described in Section 2.9. The concentrations of the CIP and PEF standard solutions (μg/spot) were chosen to inhibit the growth of *E. coli*. The FQs solutions after the oxidation reaction were used in amounts that allowed to obtain peak areas for the main degradation products comparable to the peak areas of the FQs standards; therefore, it could be assumed that the concentration of the main oxidation products on the TLC plate is similar to the concentration of the parent compounds.

On the bioautograms obtained, there were zones of inhibition of *E. coli* growth in the places of spots originating from the tested FQs, as well as their main oxidation products. The control ACVA solution applied to the plates did not inhibit bacterial growth. Pictures of an exemplary bioautogram taken in visible light and under UV light at 254 nm are shown in Figure 12.

Taking into account the above results, it can be assumed that the main oxidation products of CIP and PEF probably retained antibacterial activity against *E. coli*; however, the size of the inhibition growth zones for PEF and PEF-1, suggest that the degradation product of PEF shows less antimicrobial activity compared to the parent compound.

These conclusions are confirmed by the analysis of structure-activity relationships for the resulting oxidation products. In both cases, the structure of the quinolone ring of the degradation products of CIP and PEF remains intact after reaction with ACVA. The substituents at positions 1, 2, 3, and 4 of the degradation products CIP-1 and CIP-2 remained unchanged, which guarantees the preservation of affinity for the DNA-gyrase complex. The key to enhancing the antibacterial activity of FQs is position C-7. According to data from the literature, FQs having a five- or six-member heterocyclic ring with or without substitution at the position C-7 (pyrrolidinyl, pyrrolyl, and piperazinyl), such as CIP-1 and CIP-2, have good antibacterial activity [54]. The presence of the fluorine atom at position C-

8 significantly improves the activity of the compound against anaerobic and Gram-positive bacteria. It should thus be presumed that the CIP-2 product devoid of the fluorine atom should have less antibacterial activity, but this is not confirmed by the result obtained in the bioautography study; however, data from the literature indicate that compared to analogues without substitution at the 8 position, C-8-fluoro derivatives are more active in vivo, due to better absorption, and are only slightly less active in vitro [54,55].

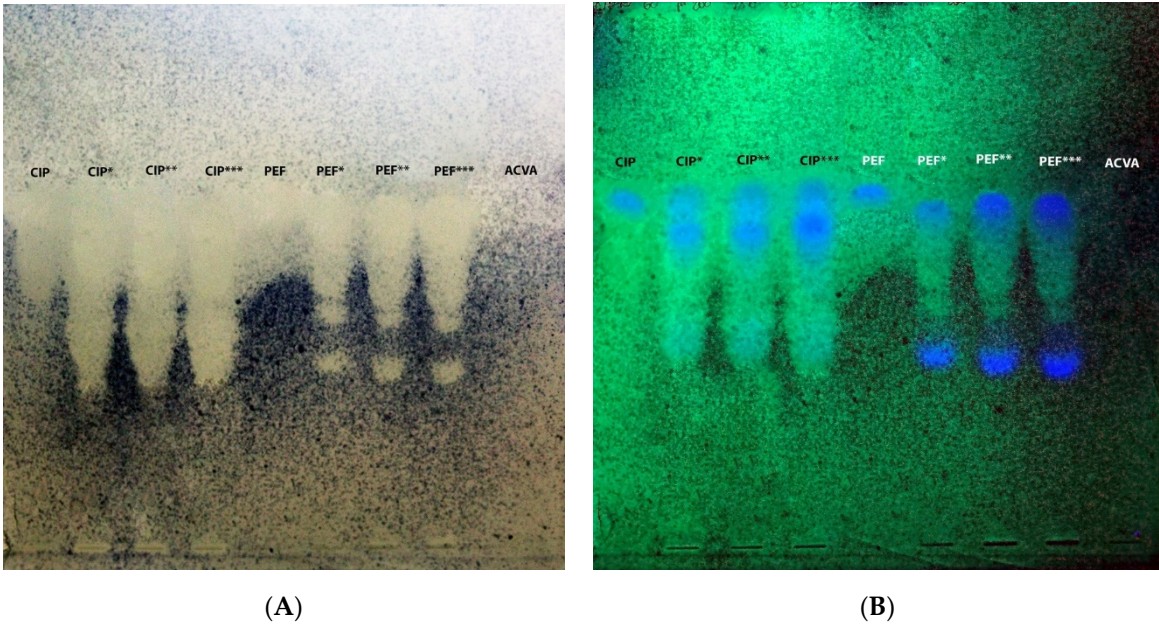

|   (A)   |   (B)   |

**Figure 12.** TLC plate sprayed with 0.2% MTT: (**A**) VIS light; white zones are an area of *E. coli* growth inhibition; (**B**) UV light 254 nm; the blue spots represent the exact places where substances were located. Track 1—standard solution of CIP, tracks 2–4—CIP sample after oxidation with ACVA, track 5—standard solution of PEF, tracks 6–8—PEF sample after oxidation with ACVA and track 9—ACVA solution after incubation at 60 °C for 60 h.

The proposed chemical structure of the PEF-1 degradation product associated with the addition of an oxygen atom to the carbonyl double bond suggests that this compound should lose its antimicrobial properties; however, the results of the TLC–DB test only indicate a reduction of the antibacterial activity against *E. coli*. Our second proposal is therefore that PEF-1 is structurally a peroxycarboxylic acid seems more likely. The PEF-1 product as a peroxycarboxylic acid retains its weak acid properties and the ability to coordinate $Mg^{2+}$ ions, thus the possibility of forming a complex with DNA gyrase. Literature data show that replacement of the carboxyl group at position C-3 with other acid groups decreases the antimicrobial activity of FQs [54].

## 4. Conclusions

The presented research is part of a project that involves the evaluation of the course of the oxidation process of selected FQs under the influence of various oxidizing agents. The results presented in this study supplement the knowledge regarding the oxidative stability of CIP and PEF. In order to assess the oxidative degradation process of the tested FQs, a new RP-HPLC-DAD method has been developed and validated. The proposed method enables the separation of CIP and PEF from their degradation products and fulfills the acceptance criteria for the stability-indicating analytical method. Investigations of the CIP and PEF oxidation process under the influence of $H_2O_2$ showed that the compounds tested were not degraded in the presence of this oxidant. The lack of decomposition of CIP and PEF under the impact of $H_2O_2$ indicates that forced oxidative degradation studies of FQs

with oxidants other than $H_2O_2$ are necessary and should be recommended for this group of drugs.

The use of the azo initiator of radical reactions, i.e., ACVA, caused the oxidative degradation of both FQs at each of the temperatures employed (40 °C, 50 °C, and 60 °C). The obtained results of the oxidation of FQs by ACVA indicate that CIP is definitely more susceptible to oxidative degradation than PEF.

The application of the UHPLC-MS/MS method allowed to determine the probable structures of the main oxidation products of CIP and PEF, as well as to propose degradation pathways. The proposed structures of CIP and PEF degradation products formed in the presence of ACVA were different from the structures of CIP and PEF oxidation products formed under the influence of $KMnO_4$ in an acidic environment described in the literature [32,35]; however, the CIP-1 product with the proposed structure was registered in a reaction of CIP with potassium permanganate at pH = 7 [43]. The other obtained oxidation products (CIP-2 and PEF-1) and their chemical structures were described in this paper for the first time.

The TLC-direct bioautography technique used and analysis of structure-activity relationships shows that the resulting CIP and PEF main oxidation products probably retained antibacterial activity against *E. coli*.

In summary, our research supplements the scientific knowledge on the oxidative stability of FQs and suggests the need to conduct research on the forced oxidation degradation of other compounds belonging to this group of drugs with the use of azo initiator of radical reactions. The obtained results are of practical importance for ensuring the appropriate quality of the finished drug forms and their storage conditions, especially since in silico studies suggest some oxidation products may show a toxic effect.

**Supplementary Materials:** The following supporting information can be downloaded at: https://www.mdpi.com/article/10.3390/pr10051022/s1, 2.8 UHPLC/MS/MS Analysis, Figure S1–S5.

**Author Contributions:** Conceptualization, U.H. and B.Ż.-W.; methodology, U.H., B.Ż.-W. and A.M.; Software, U.H., M.S. and B.Ż.-W.; validation, B.Ż.-W.; formal analysis, B.Ż.-W., P.Ż. and M.A.; investigation, B.Ż.-W. and P.Ż.; writing—original draft, B.Ż.-W., M.S. and P.Ż.; writing—review and editing, M.S. and U.H.; supervision, U.H. All authors have read and agreed to the published version of the manuscript.

**Funding:** The research was financed from the funds of the R&D project by the Polish Ministry of Education and Science N42/DBS/000156.

**Institutional Review Board Statement:** Not applicable.

**Informed Consent Statement:** Not applicable.

**Data Availability Statement:** The data are available from the authors upon reasonable request.

**Acknowledgments:** Publication was funded by the Priority Research Area qLife under the program Excellence Initiative—Research University at the Jagiellonian University in Krakow.

**Conflicts of Interest:** The authors declare no conflict of interest. The funders had no role in the design of the study; in the collection, analyses, or interpretation of data; in the writing of the manuscript, or in the decision to publish the result.

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
