# Peer review of "Study of Oxidation of Ciprofloxacin and Pefloxacin by ACVA: Identification of Degradation Products by Mass Spectrometry and Bioautographic Evaluation of Antibacterial Activity"

_processes, doi:10.3390/pr10051022_

Round 1

Reviewer 1 Report

Dear authors,

The manuscript entitled "Study of oxidation of ciprofloxacin and pefloxacin by ACVA. Identification of degradation products by mass spectrometry and bioautographic evaluation of antibacterial activity” was to to evaluate the oxidation of CIP and PEF under the influence of the azo initiator of radical reactions of ACVA. In order to achieve the above goals, new HPLC-DAD methods allowing for the determination of CIP and PEF, with its oxidation products were developed and validated. The kinetic evaluation of the oxidation process was performed and the degradation products were identified by the UHPLC- MS/MS method. It presents scientific relevance for the area of Pharmaceutical Processes.

Title: remove point from end.

Introduction - It is well written, but I suggest:

-I suggest at the end of the introduction, I suggest highlighting the "innovative" proposal of the method, as well as the advantages/disadvantages

Material

-Preparation of Samples and Execution of Oxidation Tests with the Participation of ACVA: is there any reference?

Congratulations, an interesting research article. Good results, with pertinent 
explications

Author Response

We would like to thank the reviewer for his time and valuable comments and suggestions which allowed us to improve our manuscript. 

Title: remove point from end.

Response: The point at the end of the title was removed.

Introduction - It is well written, but I suggest:

-I suggest at the end of the introduction, I suggest highlighting the "innovative" proposal of the method, as well as the advantages/disadvantages.

Response: We made some changes at the end of the introduction to emphasize the advantages and innovation of the methods used. The appropriate changes are marked in yellow.

Material

-Preparation of Samples and Execution of Oxidation Tests with the Participation of ACVA: is there any reference?

Response: As suggested by the reviewer, we have included a literature reference.

Reviewer 2 Report

My understanding of your study is that it aims to evaluate the oxidative pattern of two flouroquinolones under the influence of two oxidizing agents, H2O2 and ACVA, and to compare the suitability of these agents for being used in forced oxidative degradation studies of FQs.

In this context, could you explain in more the usefulness of the performed in silico toxicity risk assessment (lines 472-475)? Is it related either to the oxidative degradation pattern of the analysed drugs, or relevant for the ultimate goal of your research study?

Lines 517-5191 - could you please modify the phrase, in order to make sense?

Related to the Introduction section, as far as I am concerned, there are described too many aspects unrelated to the purpose of this research study (lines 30-47). Could you please shorten this section, in order to make the article more easy to read?

Line 131 - it is not clear what do you mean by "removal from the aquatic environment" in the context of PEF oxidative degradation studies. Is there an oxidative agent which needs to be mentioned?

Could you please expose the obtained results of the performed oxidative study in a more synthetic and clear manner. Which are exactly the parameters you have demonstrated to be important for the oxidative degradation of the analysed FQs and how it can be applied in drug stability studies?

Author Response

We would like to thank the reviewer for his time, valuable suggestions, and comments which allowed us to improve our work. The responses to the reviewer's comments are provided below.

My understanding of your study is that it aims to evaluate the oxidative pattern of two fluoroquinolones under the influence of two oxidizing agents, H2O2 and ACVA, and to compare the suitability of these agents for being used in forced oxidative degradation studies of FQs.

In this context, could you explain in more the usefulness of the performed in silico toxicity risk assessment (lines 472-475)? Is it related either to the oxidative degradation pattern of the analysed drugs, or relevant for the ultimate goal of your research study?

Response: Yes, our goal is to evaluate the PEF and CIP oxidation pattern under the influence of two oxidizing agents, H2O2 and ACVA, and to compare the suitability of these agents for being used in forced oxidative degradation studies of FQs. But also, if possible, we want to evaluate the resulting oxidation products in terms of their antibacterial and toxic effects. The evaluation of potential adverse reactions FQs degradation products that may appear in a drug product is very important for pharmaceutical technology and the pharmaceutical industry. Hence the idea to use a simple tool, which is the Osiris program, to assess the risk of toxic effects of the obtained oxidation products.

The section on the evaluation of the toxicity of degradation products using the Osiris program in the paper has been improved. A sentence has also been added that justifies why this type of assessment was carried out.

Lines 517-5191 - could you please modify the phrase, in order to make sense?

Response: The text has been corrected. 

Related to the Introduction section, as far as I am concerned, there are described too many aspects unrelated to the purpose of this research study (lines 30-47). Could you please shorten this section, in order to make the article more easy to read?

Response: Our idea when writing the introduction was to reach a large group of potential readers, and that is why it is so extensive, but we understand that the purpose of our work is clear without this part. As suggested by the reviewer, the indicated fragment of the text has been shortened to avoid unnecessary information.

Line 131 - it is not clear what do you mean by "removal from the aquatic environment" in the context of PEF oxidative degradation studies. Is there an oxidative agent which needs to be mentioned?

Response: The text has been corrected and information on the oxidative agent was introduced.

Could you please expose the obtained results of the performed oxidative study in a more synthetic and clear manner. Which are exactly the parameters you have demonstrated to be important for the oxidative degradation of the analysed FQs and how it can be applied in drug stability studies?

In response to the reviewer's suggestions, changes were made to the introduction, the purpose of the paper, the discussion of the results of the in silico study on the risk of toxic effects of the resulting oxidation products, and the conclusions section. All changes are marked in yellow in the text of the paper.

Round 2

Reviewer 2 Report

I would like to thank the authors for taking into consideration my suggestions related to the present research article.

All the issues found in the original paper have been addressed and properly modified. Thus, I consider that the actual paper is worth being published in Processes.